# Perspectives Associated with Human Papillomavirus Vaccination in Adults: A Qualitative Study

**DOI:** 10.3390/vaccines11040850

**Published:** 2023-04-15

**Authors:** Alina Cernasev, Kenneth C. Hohmeier, Oluwafemifola Oyedeji, Tracy Hagemann, Kristina W. Kintziger, Taylor Wisdom, Justin Gatwood

**Affiliations:** 1Department of Clinical Pharmacy and Translational Science, College of Pharmacy, University of Tennessee Health Science Center, Nashville, TN 37211, USA; acernase@uthsc.edu (A.C.); thageman@uthsc.edu (T.H.);; 2Department of Public Health, University of Tennessee Knoxville, Knoxville, TN 37996, USA; 3Department of Environmental, Agricultural & Occupational Health, College of Public Health, University of Nebraska Medical Center, Omaha, NE 68198, USA; 4US Health Outcomes Vaccines, GSK, Philadelphia, PA 19104, USA

**Keywords:** HPV vaccine, COVID-19 pandemic, vaccine hesitancy, adult population

## Abstract

Background: In the last several decades, vaccine hesitancy has become a significant global public health concern. The human papillomavirus (HPV) vaccine has been on the United States of America (USA) market since 2006, with extended approval up to age 45 granted in 2018. To date, there is limited research evaluating barriers and facilitators related to HPV vaccine initiation among adults and the influence of the COVID-19 pandemic on individuals’ vaccine-related behaviors. This study’s main objective was to characterize the contributing factors that could promote or inhibit HPV vaccine uptake for adults. Methods: A qualitative approach consisting of focus group discussions (FGDs) was used for this study. The FGD guide was informed by concepts from the Transtheoretical Model, Health Belief Model, and Social Cognitive Theory. All virtual FGDs were led by two researchers, who recorded audio for data collection. The data were transcribed by a third party, and the transcripts were imported into Dedoose^®^ software and analyzed using the six steps recommended by thematic analysis. Results: A total of 35 individuals participated in 6 focus groups over a 6-month period. Thematic analysis revealed four themes: (1) Intrinsic motivators for HPV vaccination, (2) Extrinsic motivators for HPV vaccination, (3) Vaccine promotion strategies, and (4) Impact of COVID-19 Pandemic on vaccine hesitancy. Conclusion: Both intrinsic and extrinsic factors play a role in influencing HPV vaccine uptake, and such considerations can guide efforts to improve the odds of HPV vaccination in working-age adults.

## 1. Introduction

Since the discovery of the first vaccine in 1796, vaccine hesitancy has been a significant concern in the global public health arena. In the United States of America (USA), routine immunizations are an integral part of individual care and contribute to saving lives. However, the situation changed dramatically during the COVID-19 pandemic, where vaccine hesitancy’s prevalence grew [1]. Hesitancy has been attributed to numerous factors, including healthcare disparities, concerns about safety and efficacy, health literacy, and a lack of trust in the healthcare system and providers [2,3].

The World Health Organization (WHO) defines vaccine hesitancy as a “delay in acceptance or refusal of vaccines despite availability of vaccine services. Vaccine hesitancy is complex and context-specific, varying across time, place, and vaccines. It is influenced by factors such as complacency, convenience, and confidence” [4]. Vaccine hesitancy contributes significantly to individual delay in receiving appropriate vaccines in all age groups and conditions. For example, a recent WHO report attributed vaccine hesitancy as one of the ten leading threats to global health [5]. Specific to HPV, previous studies demonstrated that parents were reluctant to allow their adolescents to receive the HPV vaccine [6,7]. Previous research attributed vaccine hesitancy to be multifactorial, including distrust in the HPV vaccine safety, preconceived mind about vaccine mechanism of action, and social media influence [8,9].

A Centers for Disease Control and Prevention (CDC) report showed that HPV contributes to more than 21,000 related cancers in women and around 15,000 cases in men annually [10]. The high prevalence of cases of HPV-related cancers contributes significantly to a vicious cycle and creates an economic burden for society [11]. The HPV vaccine was approved in 2006 in the USA for usage up to age 26. The HPV vaccine approval was expanded to adults up to age 45 in 2018 [12]. Although the USA adult population has access to an approved HPV vaccine and the coverage for adults has been expanded to 45 years old, the vaccination rate is significantly lower in adults than adolescents, and there are important gender and race differences as well [12,13]. As of 2021, 64% of female and 60% of male adolescents (ages 13–17 years) had completed the HPV series. Lower proportions of White, non-Hispanic, and Hispanic adolescents had completed the series than Black, American/Indian Alaska Native, or Asian adolescents [14]. In 2018 (most recent data available), 35% of females aged 18–26 years had completed the series, compared to 9% of males in the same age group. Further, White, non-Hispanic adults are more likely to complete the HPV series than Black or Hispanic adults [15]. A number of factors may contribute to such poor HPV vaccination rates in adults. For instance, Thompson and colleagues observed that women who have a college degree were more aware of the HPV vaccine than men [16]. In 2020, Reither and colleagues highlighted that many adult females were unaware that the recent guidelines indicated they can receive the HPV vaccine [17]. The decreased awareness among the eligible adult population might stem from poor communication skills between the provider and patient, which can hinder conveying the proper message [18]. Fontenot and colleagues emphasized the misconception perceived by men that they are not at risk of contracting the HPV infection or that HPV only produces cancer in women [19]. Subsequently, these misconceptions and beliefs might contribute to hesitancy to receive the HPV vaccine [19].

The approval of the COVID-19 vaccines has also been attributed to an increased vaccine hesitancy among the general population [1]. However, given the literature gap informing about obstacles and facilitators related to HPV initiation among adults and the influence of the COVID-19 pandemic on individuals, it is essential to assess the barriers. Thus, this manuscript aimed to examine the contributing factors that could promote or inhibit HPV vaccine hesitancy for adult individuals in light of the COVID-19 pandemic.

## 2. Methods

### 2.1. Study Design and Framework

Online focus group discussions (FGDs) were undertaken with participants from Tennessee to explore their opinions about HPV vaccine hesitancy, attitudes, and behaviors during the COVID-19 pandemic. Due to the ambiguities in the concept of vaccine hesitancy and the heterogeneity in the causes of this phenomenon, there is no widely accepted theoretical framework for this topic [20]. This study used a synergism of concept elements from the Transtheoretical Model, Health Belief Model, and Social Cognitive Theory for construct [21,22,23]. These theories support the grouping of concepts and operationalizing of definitions [24].

The FGD guide was tailored to address subjects’ perceptions and attitudes regarding HPV hesitancy during the COVID-19 pandemic and to learn how to adapt behaviors to decrease its associated hesitancy. The FGD was piloted on individuals 18–45 years of age located throughout Tennessee to facilitate a wide representation of the population. After piloting the FGD in a semi-structured manner, certain questions were modified to improve clarity. This strategy to incorporate additional questions resulted in an enhancement of the external validity of the study findings [25].

### 2.2. Subjects and Recruitment

After receiving approval from the University of Tennessee Health Science Center (UTHSC) Institutional Review Board (IRB # 21-08416-XM, approved 3 November 2021), the recruitment of subjects was initiated using a convenience sample with a snowball approach [26]. For example, fliers were placed in gyms, community colleges, yoga studios, and churches in different regions of Tennessee. The research team also contacted health clinics and community pharmacies to further facilitate the dissemination of recruitment fliers. A financial incentive in the form of a gift card was provided to participants who completed the focus groups.

The inclusion criteria to participate in this study were: (1) aged 18–45 years; (2) residing in Tennessee; (3) English speaker; and (4) willing to share their stories about vaccination in a group environment.

### 2.3. Data Collection and Analysis

The virtual FGDs were led by two researchers, and each session lasted from one to two hours for over five months in 2021 and 2022 [27]. Each FGD was audio recorded and transcribed by a professional transcriptionist. Following that, the transcripts were read by the research team, and feedback was provided in terms of saturation.

The corpus of data was imported into Dedoose^®^ (Manhattan Beach, CA, USA) software. Using thematic analysis that followed the six-step process outlined by Braun and Clarke, the data were analyzed inductively and independently by two team members [28]. After familiarization with the transcribed data, the entire dataset was coded inductively to identify emerging themes, which were reviewed by the research team; subsequently, the themes were defined and named, followed by a full written analysis [28]. The analysis team continued recruiting until thematic saturation was achieved at a point at which no new themes emerged with subsequent focus groups [25].

To ensure the rigor of qualitative research was achieved throughout the data collection, analysis, and interpretation, the team followed the steps described by Lincoln and Guba [29]. Furthermore, the research team also consulted the consolidated criteria for reporting qualitative research (COREQ) [30].

## 3. Results

A total of 35 individuals participated in six focus groups, and their characteristics are presented in Table 1. The sample was mostly female, with ages ranging from 18 to 45 years.

Four major themes were identified from thematic analysis which included: (1) Intrinsic motivators for HPV vaccination, (2) Extrinsic motivators for HPV vaccination, (3) Vaccine promotion strategies, and (4) Impact of COVID-19 pandemic on vaccine hesitancy.

### 3.1. Theme 1: Intrinsic Motivators to Use HPV Vaccine

The first theme refers to factors that are personal to an individual and describes the inherent influencers on individual vaccine perceptions. Sub-themes that emerged relating to intrinsic motivators included a lack of knowledge and information, previous negative experiences, the benefits of HPV vaccination, and personal decision/research.

#### 3.1.1. Sub-Theme 1: Lack of Knowledge and Information

A recurrent theme among several participants was inadequate knowledge about the HPV vaccine. This sub-theme describes participants’ expressions about lack of awareness and the impact of vaccine uptake. One participant expressed their view by comparing the level of information or awareness about the HPV vaccine to other recommended vaccines.

“I feel like the only vaccines we really talk about in like the news or in social media is basically the flu vaccine because it comes around every year and you have to get the flu vaccine, and now the COVID vaccine. So, I feel like other vaccines that-I know HPV vaccine is optional, but a lot of the times to get into school or work and stuff, you’re required to have like chickenpox, measles, tetanus shot, so I don’t really think that there’s a lot of information available about that vaccine right now”(FG4 P4)

In another focus group session, one participant highlighted lack of information, especially among males. The following excerpts suggest the value of vaccine promotion efforts that provide recommendations for both males and females.

“I would just want to know what it does-so I don’t even really know what it does for females, but I have no idea what it would do for males. So, I think it just needs to be explained a little bit better”(FG6 P1)

#### 3.1.2. Sub-Theme 2: Previous Negative Experience

This sub-theme represents how negative experiences about HPV infection influence perceptions about the HPV vaccine. Several participants recounted their past experiences with HPV either personally or through their friends or family. Participants discussed how their perception of risk influences their decision about the HPV vaccine. One participant recounted:

“I have had two friends that have received the HPV vaccination, and they did contract HPV and had some procedures done. And for me, I just don’t feel like I’m in that risk category. And I don’t feel like the benefits outweigh the risks, only because […] the two individuals that I do know that have received it did still get HPV.”(FG2 P1)

This view was echoed by another participant who shared their personal experience with HPV diagnosis despite receiving the HPV vaccination.

Similarly, another participant strongly expressed concern about the benefits and effectiveness of the HPV vaccine due to knowledge of HPV infection in others despite vaccination. These quotes by participants depict the importance of vaccine education that includes relevant data supporting the benefits and effectiveness of the HPV vaccine.

“But I think, even with the vaccine, it’s not effective because I just know people with the vaccine who still have HPV. It just does get spread around. [..], so I just don’t think, at that point, if people are still getting it, that the vaccine is the end all be all for what you should be getting to prevent HPV when there are other ways as well.”(FG6 P4)

Participants also noted that they tend to compare their experiences with the efficacy of routine vaccines, including the flu vaccine, for example, to their perception of the HPV vaccine. The following excerpts demonstrate that HPV vaccine hesitancy may result from experiences relating to other vaccines.

“It’s kind of why I don’t get the flu shot. The one time that I did get the flu shot, I know a lot of people say this, but the one time I did get the flu shot, I got the flu, I had strand A and B, and I was extremely sick. I haven’t gotten the flu shot in four years, and I haven’t had the flu. So, for me, I just don’t feel at risk to contract HPV, and I just don’t see the benefits in receiving the vaccination.”(FG2 P1)

Fear of side effects may discourage vaccine acceptance, especially among those who have had previous side effects from other vaccines. One participant shared their experience of having side effects following HPV vaccination and fear about side effects from the COVID vaccine:

“The same thing with the HPV vaccine, I had that, and I was in rough shape, changed my diet around, and then everything came back like really clear and good. As far as like the COVID, I was on the fence because I have a lot of underlying nerve pain and stuff, and I was scared that it would trigger it, which it did once, but I ultimately did it not for myself but for other people and for my parents, who are older.”(FG2 P4)

#### 3.1.3. Sub-Theme 3: Benefits of Receiving HPV Vaccination

A common view shared by participants is the benefit of cancer prevention. For example, the comment below highlights the participant’s belief that preventing cancer is a significant benefit of HPV vaccination. This suggests the importance of incorporating the benefits of receiving the HPV vaccine during vaccine advocacy efforts.

“I think it can benefit, aside from obviously-the one I feel is talked about most is cervical cancer in women, but I’d say everyone can benefit because where HPV is through skin-to-skin contact, and intercourse isn’t necessarily required for it to be transmitted. Obviously, avoiding cancer is a big benefit, but the benefit for everyone is to reduce transmission in the first place.”(FG5 P8)

In another discussion, participants talked about the benefit of protecting other community members in addition to themselves. The following quotation shows the importance of highlighting the benefit of protecting others, including potential partners, in future vaccine advocacy efforts.

“If I was trying to encourage somebody to get the HPV vaccine, I would just talk about how it will protect them and others from HPV infections that could potentially lead to cancers and just really drive the safety of the vaccine, the effectiveness of the vaccine, all that home and hopefully they would be more willing to get it.”(FG3 P4)

### 3.2. Theme 2: Extrinsic Motivators to Influence HPV Uptake

The second theme represents external factors related to an individual that influences opinions and perceptions about HPV vaccination. Sub-themes that emerged included healthcare provider recommendations and stigmas associated with HPV disease.

#### 3.2.1. Sub-Theme 1: Healthcare Provider Recommendations

This sub-theme revealed the common views shared by most participants regarding the importance of their health care provider’s vaccine recommendations. During all the focus group sessions, participants acknowledged that receiving recommendations about the HPV vaccine from their healthcare provider was vital in their decision to receive the vaccine or not. This underscores the value of cogent vaccine recommendations by healthcare providers. One participant shared their strong preference:

“Personally, I think at least I would be most comfortable with my primary care physician bringing that up just because I go to them with anything health related, so it just makes sense in my mind that they would talk to me about this, [..] and then we would work together on deciding if I get it or not.”(FG3 P2)

Similarly, another participant pointed out that the HPV vaccine may be perceived as unimportant if there was a lack of discussions about the HPV vaccine by healthcare providers. She comments:

“I shouldn’t have to learn about a vaccination from the TV. I feel like that’s weird and that makes me not want it because I’m going to the doctor regularly, and my doctor hadn’t mentioned it, then it must not be important.”(FG1 P5)

The following excerpts demonstrate the emphasis on the value of fostering a trusted relationship between patients and their healthcare providers. Participants outlined that patient education at the doctor’s office and the provision of relevant educational materials were helpful in vaccine decision making. However, concern about time constraints for doctors to build such relationships within the healthcare system was raised.

Another participant explained that trust in doctors’ recommendations about vaccines results from established patient-provider relationships.

In terms of timing for vaccine recommendation, the following quotation summarizes one participant’s suggestion that same-day recommendation for HPV vaccine during routine clinical visits at the doctor’s office may promote vaccine uptake.

“…I remember at least when I was a kid, everybody that I knew had a yearly physical where they’d go and see their PCP, so I think that could be a talk right there. If they would have the vaccine on hand, you could talk to the parent right there and say, your child qualifies, or even an individual going that’s over 18, by themselves, you qualify for the HPV vaccine today, would you like it. […] and then maybe go over what it is and how it could help.”(FG4 P6)

The importance in the way vaccine recommendations were communicated was also discussed in one focus group session. One participant shared how communicating with their health care provider was influential in their vaccine decision making process, suggesting that communication about vaccine recommendations should be presented in a simple and easy-to-understand way while allowing room for the individual to ask questions and seek clarification.

One obstacle to health care access mentioned by another participant was financial challenges, especially with lack of insurance. This indicates that the promotion of alternative means to cover the financial costs of vaccination may be needed for those uninsured or low-income families.

#### 3.2.2. Sub-Theme 2: Stigma and Its Negative Impact of HPV Uptake

This sub-theme represents participants’ views about the stigmas associated with HPV infection and how it influences vaccine acceptance. Participants noted the difficulty in having meaningful discussions about HPV as a sexually transmitted infection. Some participants shared that personal parental beliefs about their children not engaging in sexual activity may prevent vaccine acceptance. The following quotes illustrate the negative role of stigma and the importance of vaccine promotion messaging that encourages positive discussion about the benefits of the HPV vaccine such as cancer prevention and protecting future partners.

“But the stigma of it being a sexual thing, or people who do look at it as like, well, my kid won’t do that, so it’s not necessary, without thinking of if they have a future partner who might not have shared that particular belief system, it could still happen.”(FG5 P1)

### 3.3. Theme 3: Vaccine Promotion Strategies

This theme captures the variety of participants’ perspectives regarding vaccine promotion strategies. Most participants agreed that education is vital for increasing awareness and decreasing the lack of information about HPV vaccination.

#### 3.3.1. Sub-Theme 1: Type of Promotion

A few participants suggested HPV vaccine promotion through social media could be an option. The following extracts demonstrate the importance of utilizing social media campaigns to improve HPV vaccination among adults, particularly because teenagers and young adults can be reached through various social media outlets.

“Social media campaigns would be good since the best age to get it at is in your teens, so a lot of teenagers, they’re on social media. So having social media campaigns saying this is why you should get it and how to talk to your doctor about it would be good for them.”(FG4 P4)

Several participants recommended HPV vaccine promotion in schools. As one participant reported below, initiating health education topics about HPV vaccination within schools may prompt and encourage students to seek further information from healthcare providers or caregivers.

“I think that if it is a health risk, I think anyone who has that information should be able to give it out. […] I especially, again, at schools, if they can have a health class, they can definitely implement some kind of curriculum that shares this information.”(FG4 P4)

Several participants suggested the use of promotional materials such as pamphlets, posters, and fliers. The following comment highlights the relevance of providing information about the HPV vaccine, especially at the doctor’s office. The information presented may include the benefits, risks, and effectiveness of the HPV vaccine.

“I would say a pamphlet and a conversation with the physician explaining to me what it is, what the benefit of it is, why I should get it, and what are some downsides to it would all be very helpful for me.”(FG6 P5)

#### 3.3.2. Sub-Theme 2: Promotion Messaging

The following quote portrays participants’ view that HPV vaccine promotional messaging should also focus on the catch-up population. This will include implementing strategies to reach young adults who may have missed the opportunity to be vaccinated as an adolescent.

“The marketing pushing stuff, it is directed more at parents, […], rather than reaching out to teenagers. Also, young adults who, if their parents didn’t get them vaccinated, say, hey, it’s not too late, you can go, have your pap smears, get tested-, male or female, and you can still get the shot yourself. That needs to be something brought up more to people who are newly 18, 19, 20. […].”(FG5 P1)

One participant shared his experience with being diagnosed with HPV, suggesting that sharing a survivor’s story may motivate HPV vaccine uptake by individuals. He described his experience:

“Yes, I was just trying to say people at work sometimes will be talking about their children or a neighbor or a friend, and I will just open up and volunteer that, yes, I do have HPV, and this is what I’ve been through, and if I had had the choice, I would have taken it. That way they get it from a real human being they know rather than a doctor or a piece of paper.”(FG2 P5)

To promote adherence to vaccine appointments, one participant suggested a quicker appointment for vaccination to decrease the time commitment that an appointment can take. This is particularly important for individuals who may consider time constraints as a barrier to vaccination.

“I think that speed, going to the office and just kind of in and out would be better because I know a lot of times like appointments, you’ll go in, and then they’ll be like, just have a seat, and you’ll be waiting there for 15–20 min. So, you know, you’re spending almost an hour just to get another shot.”(FG3 P5)

#### 3.3.3. Sub-Theme 3: Promoting Adherence to Vaccine Schedules

Another suggestion to improve adherence to vaccination was related to reminder calls for follow-up appointments. Many participants discussed the value of reminders about their vaccine appointment from their doctor’s office.

Additionally, the following excerpt illustrates participants’ preference for phone calls about vaccination. This is important to engage patients in conversations that will help address questions and concerns about vaccination.

“As far as receiving follow-up when you receive a vaccination, or if you’re trying to inform me about a vaccination, a text message is just not appropriate because, you know, people may have questions. It’s the same thing as advertising vaccines on television versus having your PCP […] explaining it to you because it leaves room for discussion. A text message, most of the times, those automated messages from the doctor are not- you can’t respond.”(FG1 P5)

### 3.4. Theme 4: Impact of COVID-19 Pandemic on Vaccine Hesitancy

Across all focus group sessions, conversations about COVID-19 vaccines were frequently discussed. These discussions portrayed the impact of COVID-19 on perceptions about vaccinations in general. However, it is important to note that the sense of urgency to be vaccinated against diseases with long-term complications such as HPV-related cancers may be delayed compared to diseases with immediate complications like the coronavirus infection.

“For me, personally, it has made me a lot more willing to get every other vaccine because like with HPV, that’s not something that we’re seeing the effects of right away like with COVID, and like just seeing how many people have died from it, that has made me see the importance in it a lot more.”(FG3 P5)

Other participants shared that if family and friends had experienced severe disease from COVID-19, then that could influence their decision to be vaccinated. This suggests that sharing experiences about the severity of the disease may prompt individuals to receive vaccines.

“Yeah, definitely, it makes me more willing to get vaccines. Certainly when, my sister had asthma, she got the vaccine, she got both of them, and she didn’t think she was going to get a booster, and then she ended up getting COVID, and she wouldn’t know where she was without the vaccine because it was so bad even with it.”(FG3 P5)

Another participant expressed hesitancy about the COVID-19 vaccine due to their health status and lack of knowledge about the components of the vaccine.

One participant indicated that the promotion of other vaccines with COVID 19, including the HPV vaccine may have a negative impact on vaccine acceptance.

A few of the participants discussed the impact of politics on vaccination in recent times suggesting that the politicization of vaccination may fuel vaccine hesitancy.

“I grew up getting vaccinations, but I am a part of the group that is a little bit hesitant about vaccinations…But I do feel like vaccines are a little bit too politicized nowadays. I do feel like a lot of the drawback from vaccinations is I don’t feel like my parents’ generation had as much access to information the way we do”(FG2 P1)

## 4. Discussion

This qualitative research study used focus groups across Tennessee to explore adult perceptions of the HPV vaccine and contributing factors that promote or inhibit HPV vaccination. These exploratory findings are especially important in lieu of a recent expansion in vaccine eligibility for patients up to age 45 [12]. Our results shed light on patient perspectives on how the COVID-19 pandemic impacted views of the HPV vaccine, uncovered what motivates adults to uptake the HPV vaccine, and described HPV vaccine promotion strategies.

COVID-19 has had a significant impact on vaccine-related provider and patient behaviors [31,32,33]. Perhaps the most obvious impact has been in increasing awareness of the idea of vaccine hesitancy, and its role as a significant barrier to vaccination in the US and abroad [33,34]. Although declining overall vaccination rates have been reported in some studies, a recent meta-analysis of 27 pandemic-era studies (January 2019–December 2021) found there was increased intention to vaccinate across age, sex, and occupation [31,32]. This effect was also seen in parental willingness to consent to vaccination [35]. Our study indicates similarly held views among adults in Tennessee, with participants noting that seeing the effects of COVID-19 in the population at large and within their own social networks made them more cognizant of the dangers of infectious diseases and potential benefits of immunization.

Aside from COVID-19, several other motivators for HPV vaccination were described by participants during the focus groups. Subjects noted the importance of increasing the availability of information about the vaccine given that it is not openly discussed as frequently as other vaccines within the media or in conversations with family or friends. Male participants noted that HPV vaccines are promoted and viewed as a “female vaccine,” and so it is not viewed as a health priority for them. This is reflected in the literature, too, with a recent survey finding that women were more likely to have been recommended the vaccine by a healthcare provider and that only 15–30% of participants knew it caused other cancers beyond cervical cancer [36]. In other cases, it was the knowledge of a negative experience with the HPV vaccine that impacted HPV vaccine perceptions. Participants noted that hearing of individuals contracting HPV even after having the vaccine negatively influenced their willingness to be vaccinated, and this phenomenon has also been highlighted in the published literature on HPV vaccination [37].

Participants noted that some messaging positively impacted vaccination motivation. This included patient knowledge of other routes of transmission, and that HPV is not solely a sexually transmitted infection (STI). The de-stigmatization of the vaccine may indeed follow from this line of thinking, as an STI is associated with moral choice, whereas a generally infectious disease is associated with probability [38]. Other routes of transmission for HPV include fomites, fingers, mouth, and skin [39]. Moreover, the idea of being vaccinated not for oneself but more broadly for the larger community resonated with participants. This idea of protecting one’s community, family, and peers is seen in other vaccines, such as the COVID vaccine [40,41]. These findings overlap with those of the WHO hesitancy framework-especially complacency and confidence [4].

Like other studies, healthcare provider recommendations were noted to be an important factor in the willingness to be vaccinated. Participants felt that because of the sensitive nature of the conversation, a recommendation from a trusted healthcare provider with which they had established rapport made the most sense. Importantly, it was noted that although awareness of the vaccine may come from other sources, the absence of a provider recommendation signaled to the patient that “it must not be important.” The way that the recommendation was provided was noted to be of high importance. Some participants, who were initially hesitant after an initial recommendation also described the critical need for shared decision making. Others noted that clear, concise recommendations during a routine visit would be preferred. This too is supported by previously published evidence which demonstrated that HPV vaccine recommendations made by pediatricians in a “presumptive” format, along with shared decision making and motivational interviewing approaches, is more likely to lead to acceptance of the recommendation [42]. This combined “presumptive” vaccine recommendation with a shared decision-making model was also tested in the community pharmacy setting with pneumococcal vaccines and found to lead to higher vaccination rates as compared with control [43]. Participants also noted that the multi-dose series of vaccination posed a barrier due to lack of convenience, but that providers taking responsibility for improving vaccine convenience such as reducing wait times and reminding patients to complete their vaccine series would be helpful. Provider “nudges” within clinical or pharmacy software which remind the clinician to make recommendations to complete vaccine series have been shown helpful and may be useful in improving HPV vaccine rates as well [44,45].

Overall, this study outlined several directions for promoting HPV vaccinations to eligible adults. Messaging around the vaccine should be such that it is a gender-neutral vaccine, rather than being a “female vaccine.” Similarly, promotion should present this as an infectious disease, public health vaccination, being careful to avoid its sole association with STI prevention. Benefits to both the individual but also to the community at large should be explained as these self-interest and community interests are not mutually exclusive [41]. Such public awareness campaigns are valuable based on results from this study, but only when combined with provider recommendations to emphasize its unique importance to the individual. These recommendations should be clear and concise with carefully selected language (i.e., “presumptive” format), using shared decision making if vaccine hesitancy is encountered [42,43]. Finally, patients value convenience in vaccine access given the multi-dose nature of the series, and vaccine providers should create solutions to ensure quick vaccination visits with reminders for the patient to complete their vaccination series.

## 5. Strength and Limitations

The qualitative study design enabled the research team to examine the personal experiences of participants regarding HPV vaccine hesitancy in-depth during the COVID-19 pandemic that would have otherwise been missed using a quantitative design.

This is the first study to utilize a heterogeneous sample of the Tennessean population regarding race, ethnicity, urban, and rural. However, the results of this study cannot be generalizable to all adults due to the qualitative research approach. Furthermore, the population was limited to residents of the state of Tennessee. It would be valuable to follow this study up with further research where the sample could be selective and focus on specific ethnicities and rural areas.

## 6. Conclusions

Patient behaviors toward HPV vaccination are influenced by a unique set of attitudes, beliefs, and behaviors. Both intrinsic and extrinsic factors play a role in influencing HPV vaccine uptake, including reducing overall cancer rates, stigma, past experiences, and convenience. Several vaccine promotion strategies were perceived to positively influence HPV vaccine seeking behaviors.

## Figures and Tables

**Table 1 vaccines-11-00850-t001:** Participant demographics.

Sex	n (%)
M	12 (33%)
F	23 (66%)
**Race**	
African American	10 (29%)
Asian	1 (3%)
White	24 (68%)
**Region**	
East Tennessee	14 (40%)
Middle Tennessee	11 (31%)
West Tennessee	10 (29%)
**Age**	
18–22	6 (17%)
23–29	15 (43%)
30–35	7 (20%)
36–45	7 (20%)

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
