# Peer review of "Perspectives Associated with Human Papillomavirus Vaccination in Adults: A Qualitative Study"

_vaccines, 2023, doi:10.3390/vaccines11040850_

Round 1

Reviewer 1 Report (Previous Reviewer 3)

Reviewer Comments:

General: A qualitative focus group discussion approach was used to assess factors that may influence HPV vaccine uptake.  Thematic analysis revealed 4 important themes.  Results are presented in a qualitative manner and could be better related to known factors associated with vaccine hesitancy in the discussion.

Title: The title “Factors associated with human papillomavirus vaccination in adults: a qualitative study” is very general implies that specific factors associated with actual vaccination will be provided in the study.  The title leads the reader to expect that the study may include vaccinated and non-vaccinated participants.  However, the authors do not specify whether the study participants have received an HPV vaccination or not.  I suggest replacing ‘Factors’ with ‘Opinions’ or ‘Perspectives’. 

Abstract:

Lines 15-18: The abstract summarizes the 4 identified themes in a different order than the results section.  The results section starts with intrinsic motivators to use HPV vaccine, and logically follows with extrinsic motivators and vaccine promotion strategies.  The impact of COVID-19 on vaccine hesitancy overall is more general as presented in the results, and although important, does not seem like it should be listed first.

Introduction:

Generally nicely written, however, the first sentence (line 24-25) about the discovery of the first vaccine in 1796 is a bit out of place.  It could be omitted.

It would be worth mentioning the specific population coverage percentages for HPV vaccination for adolescents and adults, as well as sex and race specific rates in general.  Even though the study does not report what percentage of their participants have been vaccinated, it would be useful to know what the HPV vaccination uptake in general is in order to better understand the overall acceptance of the HPV vaccine that the authors are addressing. 

Methods:

Lines 70-72: “The foci for this study were comprised by the challenging phenomenon of vaccine hesitance in addition to the fact that there is no theoretical framework for this topic.”  This is an awkward sentence that should be reworded. In the introduction, the authors mention that the WHO definition which comprises complacency, convenience, and confidence as a general framework.  There is also the WHO SAGE Working Group Vaccine Hesitancy Scale.  Since vaccine hesitancy is raised as a topic in the introduction, about how the author’s qualitative findings relate to these measures.  For example, how did their findings relate to know factors associated with vaccine hesitancy.

If possible, a better description of why a population in Tennessee was chosen.  Was this out of convenience, or is there a specific reason people in Tennessee would be of interest for HPV vaccine or vaccines in general?  Since COVID-19 vaccination is mentioned, it might be worth mentioning COVID-19 vaccination rates in Tennessee vs. the rest of the country (e.g., similar, lower, higher).  Opinions on HPV vaccine may differ in a state with higher COVID-19 vaccination coverage, and opinions on HPV in Tennessee might be of particular interest because COVID-19 vaccination coverage is lower.

Results:

The results would be easier to read if:

1.    Direct quotes from participants were offset by indentation, which would give the reader the choice to skim through them.  As it is currently presented, the reader is somewhat forced to read every word and the authors’ key points tend to get lost in the detail.

2.    Lines 120-121: Mistrust of the medical community, and personal decision/research are mentioned here but not addressed in any further detail.  If they emerged as important opinions, they should be described in a bit more detail.  If they are not important, perhaps delete them as it leaves the reader looking for more information.

3.    Lines 196-197: This is labelled ‘Theme 2’ but then the sentence starts with “The third theme represents…”

4.    Theme 3 organization could be improved, and could potentially have subthemes or subtopics for example could be type of promotion, messaging of promotion, and promoting adherence.  It may be worth mentioning that the vaccine requires two visits to clarify why adherence to appointments matters.  This is different from several other vaccinations which can be completed at a one-time visit.

5.    Theme 4 could also have subtopics.  While this theme is important, the mention of mandates (342-348) may have relevance to COVID-19 vaccination as a pandemic situation, but diverges from the focus of the manuscript since it not as generally applicable to HPV vaccines.

Discussion

Generally well written, but should pull in a few sentences or a paragraph to address known factors associated with vaccine hesitance as noted in the review of methods above.  For example, how do the findings relate to the WHO definition of vaccine hesitancy comprised of complacency, convenience, and confidence as a general framework.  The lower generalizability (438-439) is mentioned as a limitation and future directions are mentioned, but reasons for focusing on specific ethnicities or rural areas for HPV vaccination in general are not specified. 

Author Response

Reviewer 1

General: A qualitative focus group discussion approach was used to assess factors that may influence HPV vaccine uptake.  Thematic analysis revealed 4 important themes.  Results are presented in a qualitative manner and could be better related to known factors associated with vaccine hesitancy in the discussion.

Title: The title “Factors associated with human papillomavirus vaccination in adults: a qualitative study” is very general implies that specific factors associated with actual vaccination will be provided in the study.  The title leads the reader to expect that the study may include vaccinated and non-vaccinated participants.  However, the authors do not specify whether the study participants have received an HPV vaccination or not.  I suggest replacing ‘Factors’ with ‘Opinions’ or ‘Perspectives’. 

Response: Thank you for your valuable feedback. We incorporated it into the title.

Abstract:

Lines 15-18: The abstract summarizes the 4 identified themes in a different order than the results section.  The results section starts with intrinsic motivators to use HPV vaccine, and logically follows with extrinsic motivators and vaccine promotion strategies.  The impact of COVID-19 on vaccine hesitancy overall is more general as presented in the results, and although important, does not seem like it should be listed first.

Response: Thank you for the suggestion. We have revised the abstract accordingly.

Introduction:

Generally nicely written, however, the first sentence (line 24-25) about the discovery of the first vaccine in 1796 is a bit out of place.  It could be omitted.

Response: We value your suggestion. However, the intention of this sentence was to inform the reader that the vaccine hesitancy has been a concern for over a century.

It would be worth mentioning the specific population coverage percentages for HPV vaccination for adolescents and adults, as well as sex and race specific rates in general.  Even though the study does not report what percentage of their participants have been vaccinated, it would be useful to know what the HPV vaccination uptake in general is in order to better understand the overall acceptance of the HPV vaccine that the authors are addressing. 

Response: We have added a section in the Introduction, lines 59-66, to include this information.

Methods:

Lines 70-72: “The foci for this study were comprised by the challenging phenomenon of vaccine hesitance in addition to the fact that there is no theoretical framework for this topic.”  This is an awkward sentence that should be reworded. In the introduction, the authors mention that the WHO definition which comprises complacency, convenience, and confidence as a general framework.  There is also the WHO SAGE Working Group Vaccine Hesitancy Scale.  Since vaccine hesitancy is raised as a topic in the introduction, about how the author’s qualitative findings relate to these measures.  For example, how did their findings relate to know factors associated with vaccine hesitancy.

Response: We appreciate the suggestion and have edited this sentence to provide clarity.

If possible, a better description of why a population in Tennessee was chosen.  Was this out of convenience, or is there a specific reason people in Tennessee would be of interest for HPV vaccine or vaccines in general?  Since COVID-19 vaccination is mentioned, it might be worth mentioning COVID-19 vaccination rates in Tennessee vs. the rest of the country (e.g., similar, lower, higher).  Opinions on HPV vaccine may differ in a state with higher COVID-19 vaccination coverage, and opinions on HPV in Tennessee might be of particular interest because COVID-19 vaccination coverage is lower.

Response: Thank you for this suggestion. We have included verbiage under “Subjects and Recruitment” to clarify this was a convenience sample. We agree with the reviewers comment on the impact of COVID-19 on the results. However, because the sample is qualitative and the aim was saturation (rather than representation of the sample) it would be difficult to draw as clear a line between general population trends in Tennessee and our specific sample.

Results:

The results would be easier to read if:

  1. Direct quotes from participants were offset by indentation, which would give the reader the choice to skim through them.  As it is currently presented, the reader is somewhat forced to read every word and the authors’ key points tend to get lost in the detail.

Response: Thank you for this suggestion. We indented all the quotes. The original manuscript had more spaces between quotes for this purpose, however, the journal has specific rules about spacing and it was removed in the editorial process.

  1. Lines 120-121: Mistrust of the medical community, and personal decision/research are mentioned here but not addressed in any further detail.  If they emerged as important opinions, they should be described in a bit more detail.  If they are not important, perhaps delete them as it leaves the reader looking for more information.

Response:  We value your opinion. We removed the “mistrust” from the results as we did not go into further details.

  1. Lines 196-197: This is labelled ‘Theme 2’ but then the sentence starts with “The third theme represents…”

Response:  Thank you for letting us know. We amended the manuscript.

  1. Theme 3 organization could be improved, and could potentially have subthemes or subtopics for example could be type of promotion, messaging of promotion, and promoting adherence.  It may be worth mentioning that the vaccine requires two visits to clarify why adherence to appointments matters.  This is different from several other vaccinations which can be completed at a one-time visit.

Response:  Thank you for letting us know. We amended the manuscript to include 3 new sub themes as recommended.

  1. Theme 4 could also have subtopics.  While this theme is important, the mention of mandates (342-348) may have relevance to COVID-19 vaccination as a pandemic situation, but diverges from the focus of the manuscript since it not as generally applicable to HPV vaccines.

Response: Thank you for this valuable recommendation regarding the mandates. We decided to remove the information and the quote as it was diverging from the focus of the manuscript.

Discussion

Generally well written, but should pull in a few sentences or a paragraph to address known factors associated with vaccine hesitance as noted in the review of methods above.  For example, how do the findings relate to the WHO definition of vaccine hesitancy comprised of complacency, convenience, and confidence as a general framework.  The lower generalizability (438-439) is mentioned as a limitation and future directions are mentioned, but reasons for focusing on specific ethnicities or rural areas for HPV vaccination in general are not specified. 

Response: Thank you for this suggestion. We’ve gone ahead and connected the WHO framework to the discussion.

Reviewer 2 Report (New Reviewer)

The authors show the results of the study "Factors associated with human papillomavirus vaccination in 2 adults: a qualitative study " and discuss some reason for people who have hesitancy for HPV vaccination.

The study shows the opinions of only 35 participants, I consider that the sample is to small, and is circumscribed only to Tennessee.

The authors try to relate the hesitance about HPV vaccination and the situation of the Covid-19 pandemic, however, they do not comment on how these doubts about vaccination have changed through times not associated with the pandemic.

It would have been interesting for the authors to discuss how these types of hesitance about vaccination have changed over time, or whether they remain the same. Also, the study is limited to just one region with participants who may have their own idiosyncrasies, and not reflect what people may think across the country or the world.

I also believe that the Methodology section should better explain the methods for the analysis of the results.

The results section should better describe the data obtained and explain them as tables and/or graphs, since they only mention "one participant said...", but what about the rest, did they think the same?, do they have their own motivations?, etc.

The table presented in the results section is incomplete, or at least not clear, it mentions 23 participating women, but does not include the 12 men who also participated, the rest of the data is based on the 35 participants, but it would have been interestingly, each of the analyzed characteristics, such as age, ethnicity, etc., had also been represented by sex.

Author Response

The authors show the results of the study "Factors associated with human papillomavirus vaccination in 2 adults: a qualitative study " and discuss some reason for people who have hesitancy for HPV vaccination.

The study shows the opinions of only 35 participants, I consider that the sample is to small, and is circumscribed only to Tennessee.

Response:  Thank you for your suggestion. However, this is a qualitative study where the sample size is not determined by power. In contrast to quantitative studies that depend on statistical power to detect differences and use a statistical formula to determine this sample size, qualitative research uses “theoretical saturation”, which is the standard in qualitative methods. To achieve this theoretical saturation, the research team recruited participants until the data (obtained from Focus Groups) provided no additional information on the topic. In other words, sample size is not set a priori as in quantitative studies, but is driven by the research process.

For example, Sandelowski (1995, p. 179) stated that ‘There are no computations or power analyses that can be done in qualitative research to determine a priori the minimum number […] of sampling units required.”

Here are some references that we followed:

Guest, G.; Bunce, A.; Johnson, L. How many interviews are enough? An experiment with data saturation and variability. Field Methods 2006, 18, 59–82.

Archibald, M.M.; Ambagtsheer, R.C.; Casey, M.G.; Lawless, M. Using Zoom videoconferencing for qualitative data collection:Perceptions and experiences of researchers and participants. Int. J. Qual. Methods 2019, 18, 1609406919874596.

Braun, V.; Clarke, V. Using thematic analysis in psychology. Qual. Res. Psychol. 2006, 3, 77–101.

Loh, J. Inquiry into issues of trustworthiness and quality in narrative studies: A perspective. Qual. Rep. 2013, 18, 1.

Yardley, L. Dilemmas in qualitative health research. Psychol. Health 2000, 15, 215–228.

It would have been interesting for the authors to discuss how these types of hesitance about vaccination have changed over time, or whether they remain the same. Also, the study is limited to just one region with participants who may have their own idiosyncrasies, and not reflect what people may think across the country or the world.

Response: Thank you for this clarification. We mentioned this point in our limitation, which is a lack of generalizability for any qualitative study. The purpose of qualitative research is to provide in-depth understanding and explanations of a topic or phenomenon rather than to generalize findings.

I also believe that the Methodology section should better explain the methods for the analysis of the results.

Response: Thank you for this clarification. We followed the Thematic Analysis and the 6 steps described by Braun and Clarke. “Similar codes were grouped into categories. All the categories were clustered and analyzed to uncover the major themes.” 

The results section should better describe the data obtained and explain them as tables and/or graphs, since they only mention "one participant said...", but what about the rest, did they think the same?, do they have their own motivations?, etc.

Response:  We value your opinion.  However, the main objective of qualitative studies is to ensure that the sample size is small enough to capture “a new and richly textured understanding of experience.” Sandelowski, M. (1995).

Sandelowski, M. (1995). Sample size in qualitative research. Research in Nursing & Health,18, 179–183.

The table presented in the results section is incomplete, or at least not clear, it mentions 23 participating women, but does not include the 12 men who also participated, the rest of the data is based on the 35 participants, but it would have been interestingly, each of the analyzed characteristics, such as age, ethnicity, etc., had also been represented by sex.

Response: Thank you for this note. We’ve included male sex as a category within the table and added both the count and percentage.

Round 2

Reviewer 1 Report (Previous Reviewer 3)

The authors nicely responded to comments and the manuscript is improved and reads more clearly.  Thank you.

Reviewer 2 Report (New Reviewer)

After reviewing the changes, corrections, and responses, I consider the article could be considered for publication.

This manuscript is a resubmission of an earlier submission. The following is a list of the peer review reports and author responses from that submission.

Round 1

Reviewer 1 Report

1. The paper provides anecdotal evidence of how people might have vaccine hesitancy about the HPV vaccine. The modality is several focus groups in the state of Tennessee. The focus group discussions were transcribed and the authors use a text-based tool to extract themes from the transcripts. The authors then provide a taxonomy of anecdotal claims about why people may have vaccine hesitancy or might not. The authors conclude that the discussion, although limited to 36 people, can provide valuable insight to policy makers.  

2. Strength: It appears to be the only attempt to use focus group discussion applied to HPV vaccine hesitancy. The study does have interesting quotes from those who partiicpated. There does appear to be a taxonomy of sorts that comes out of the transcripts.    Weakness: Small sample size, anecdotal evidence, hard to confirm, sample is from a single state and from a single time period. There is nothing the authors can do about these weaknesses ex post.   

3. I have no major recommendations to change the manuscript. Perhaps make sure that there are no typographical errors?  

4. Minor comments: there is one typographical error on the first page and I thought I had seen another typographical error on another page but I cannot find it when I go back to the paper.    Overall, the paper is well written and the methodology is clearly outlined. The text-based tool is novel to me but appears to be used in other studies and the taxonomy that comes out of the text analysis appears reasonable. Whether there could be other points that come out of the transcripts that the authors do not report is untestable. 

Author Response

Reviewer # 1

  1. The paper provides anecdotal evidence of how people might have vaccine hesitancy about the HPV vaccine. The modality is several focus groups in the state of Tennessee. The focus group discussions were transcribed and the authors use a text-based tool to extract themes from the transcripts. The authors then provide a taxonomy of anecdotal claims about why people may have vaccine hesitancy or might not. The authors conclude that the discussion, although limited to 36 people, can provide valuable insight to policy makers.  
  2. Strength: It appears to be the only attempt to use focus group discussion applied to HPV vaccine hesitancy. The study does have interesting quotes from those who partiicpated. There does appear to be a taxonomy of sorts that comes out of the transcripts.    Weakness: Small sample size, anecdotal evidence, hard to confirm, sample is from a single state and from a single time period. There is nothing the authors can do about these weaknesses ex post.   
  3. I have no major recommendations to change the manuscript. Perhaps make sure that there are no typographical errors?  

Response: Thank you for your time to review our manuscript. The typos were addressed.

  1. Minor comments: there is one typographical error on the first page and I thought I had seen another typographical error on another page but I cannot find it when I go back to the paper.    Overall, the paper is well written and the methodology is clearly outlined. The text-based tool is novel to me but appears to be used in other studies and the taxonomy that comes out of the text analysis appears reasonable. Whether there could be other points that come out of the transcripts that the authors do not report is untestable. 

Thank you - we've also reviewed for typographical errors.

Reviewer 2 Report

The study merits attention as it is a good attempt at identifying the various factors perceived to be important in determining vaccine acceptance among volunteers. One aspect that I found interesting is the covid 19 vaccination willingness and its correlation to the nonacceptance of the HPV vaccine.  It seems that perceptions regarding vaccine efficacy have a major bearing on the decision to get vaccinated. The advent of covid 19 and the vaccines thereof played a significant role in vaccine decision-making. There needs to be a detailed study examining the inducers regarding vaccine acceptance and hesitancy to covid vaccines along with the possible sequelae for other vaccines. The earlier concept of a vaccine as a prophylactic that could protect for a  determinate time has changed and is now an event that is guided more by necessity rather than choice. It would be worthwhile to study the aspects of vaccine acceptance after the covid 19 pandemic. There could be instances of reluctance due to theories that are conspiratorial and discouraging to vaccine acceptance. The earlier perception of vaccines was more stable and robust when compared to the present day. Studies regarding the shifts in perception will be enlightening. However, the present study has one caveat in terms of the ethnicities, especially if the break up is more homogenous and uniform better conclusions can emerge.

Author Response

The study merits attention as it is a good attempt at identifying the various factors perceived to be important in determining vaccine acceptance among volunteers. One aspect that I found interesting is the covid 19 vaccination willingness and its correlation to the nonacceptance of the HPV vaccine.  It seems that perceptions regarding vaccine efficacy have a major bearing on the decision to get vaccinated. The advent of covid 19 and the vaccines thereof played a significant role in vaccine decision-making. There needs to be a detailed study examining the inducers regarding vaccine acceptance and hesitancy to covid vaccines along with the possible sequelae for other vaccines. The earlier concept of a vaccine as a prophylactic that could protect for a  determinate time has changed and is now an event that is guided more by necessity rather than choice. It would be worthwhile to study the aspects of vaccine acceptance after the covid 19 pandemic. There could be instances of reluctance due to theories that are conspiratorial and discouraging to vaccine acceptance. The earlier perception of vaccines was more stable and robust when compared to the present day. Studies regarding the shifts in perception will be enlightening. However, the present study has one caveat in terms of the ethnicities, especially if the break up is more homogenous and uniform better conclusions can emerge.

Response: Thank you for your time to review our manuscript. Ethnicities were not captured as part of the interviews. Sex, Race, Region, and Age are all reported in Table 1.

Reviewer 3 Report

The topic of the manuscript is interesting and the time taken to obtain the information is acknowledged.  The manuscript has much room for improvement in all sections as noted in the comments document.  Full details are summarized in the attached Word document and key issues are:

1. Vaccine hesitancy should be defined and HPV epidemiology should be more clearly stated.  

2. Demographics should be expanded if at all possible (as detailed in attached file)

3. In the current format, it is difficult to determine the key points from the results section amidst the details.  Reading selected individual participant comments, while illustrative and instructive, makes for a difficult read.  Some suggestions are in the attached document.

Detailed content from uploaded Word file titled '31Jan2023 Review' is as follows:

Abstract

Please clarify:

Line 15

United States of America is a little awkward, more often I see United States (US) in manuscripts, unless USA is a journal standard.

Line 17

‘the influence of the COVID-19 pandemic on individuals [what?]’

Do you mean…

Vaccination behaviors or opinions. 

Could rephrase to make it more clear.

Line 25

Vaccine hesitancy is used in line 13; vaccination hesitancy in line 25.  

Perhaps stick with one term.

Introduction

First paragraph

Line 33

The first sentence is an unusual start to an introduction.  A sentence stating the current state of vaccine hesitance would be more meaningful.  For example, what percentage of the current population could be considered vaccine hesitant, or if unknown, what percent do not get vaccinated.  

Is not getting vaccinated the same as vaccine hesitancy?  A better explanation is needed for the reader here with a definition of vaccine hesitancy and a reference.  

Line 38-39 has an extra comma

Sentence 39 makes this point better about vaccine hesitancy, and might make a good starting sentence if it included a definition, some statistics, and a reference.

Line 50

When was the HPV vaccine coverage expanded to 45 years compared to its availability in children?  What are the specific ACIP recommendations for this vaccine in adults?

Did coverage for adults happen during the pandemic when people were not getting well visits in person? While it is mentioned in the discussion that intention to vaccinate may have increased, vaccination rates in general could tend to decline during the pandemic, particularly if it is a multi-vaccine series administered in person.

Line 204

Indentation is off.

Methods

I do not see that the time period of result collection is mentioned, although it looks like it was IRB approved in November 2021.  It is difficult to relate influence of the COVID-19 pandemic to the participant comments when it isn’t clearly stated that it is presumably after COVID-19 vaccines became available.  

It would also be interesting to know why specifically people in Tennessee were chosen.  Are there any known aspects about vaccination rates in Tennessee that would make this population of interest?  For example, average vaccination rates, high vaccination rates, low vaccination rates compared to other parts of the country.

Results

While the workload of this study seemed labor intensive, participant demographics are very general.  I am left wondering more about who these people are to know who their responses might apply to.  For example, any of the following would be helpful:

·      Have they themselves been vaccinated?  

·      Are they sexually active?  Monogamous?  Married? 

·      Do they have children who they have agreed to have vaccinated?  

·      What is their level of education?  Some are of college age, are they in college?  

·      Going a step further, have they had HPV?  Has their doctor recommended HPV vaccine for them?  

·      Are they people who have received other vaccines (e.g., influenza, COVID, etc.?). 

Overall, it is very difficult to tell who these general results apply to.  

Also, for men vs. women, women might be more likely to see a doctor (gynecologist) for routine care whereas men ages 18-45 might not be seeing a doctor for routine care.  None of this information is provided.  

I do not know the relevance of East, Middle and West Tennessee as reported in the table.

As a clinician reading the results, I do not find them helpful.  They are detailed for the sake of completeness, but it is tedious and time consuming to read through the individual’s exact words without a summary table of what the gist of the content is, or what can be done with this information.  

Perhaps the participants’ exact words could be put in a supplemental section or offset from the main text so the reader could skip over if they don’t want to read the comments verbatim.

The results read as if they would be of interest to industry to sell a product as opposed to being useful to a clinician recommending a vaccine for a patient.

Discussion

The discussion nicely summarizes the results and puts them in context.  A few points that could be considered mentioning to put the results in additional context:

1.     More information on demographics of who the people surveyed are and/or who their opinions might represent (see above).

2.     Information on why this broad age range was chosen, aside from the vaccine being approved for this age group.  

3.     From a public health perspective, it would be interesting to know the opinions of the group at highest risk for new infections that could be prevented by the vaccine.

4.     Are there HPV messaging campaigns?  What information already exists that these participants may have access to?  It mentions that HPV vaccines are ‘promoted and viewed’ as female vaccines.  I wonder how the vaccines might have been promoted and how people have access to the information.

The final paragraph is a nice summary of the results and their importance.

Author Response

Reviewer #3

The topic of the manuscript is interesting and the time taken to obtain the information is acknowledged.  The manuscript has much room for improvement in all sections as noted in the comments document.  Full details are summarized in the attached Word document and key issues are:

  1. Vaccine hesitancy should be defined and HPV epidemiology should be more clearly stated.  
  2. Demographics should be expanded if at all possible (as detailed in attached file)
  3. In the current format, it is difficult to determine the key points from the results section amidst the details.  Reading selected individual participant comments, while illustrative and instructive, makes for a difficult read.  Some suggestions are in the attached document.

Detailed content from uploaded Word file titled '31Jan2023 Review' is as follows:

Abstract

Please clarify:

Line 15

United States of America is a little awkward, more often I see United States (US) in manuscripts, unless USA is a journal standard.

Response: We used the USA versus the US because the journal has an international audience.

Line 17

‘the influence of the COVID-19 pandemic on individuals [what?]’

Do you mean…

Vaccination behaviors or opinions. 

Could rephrase to make it more clear.

Response: Added phrase to improve clarity.

Line 25

Vaccine hesitancy is used in line 13; vaccination hesitancy in line 25.  

Perhaps stick with one term.

Response: Thank you for your suggestion. Therefore, we amended the text and used “vaccine hesitancy.”

Introduction

First paragraph

Line 33

The first sentence is an unusual start to an introduction.  A sentence stating the current state of vaccine hesitance would be more meaningful.  For example, what percentage of the current population could be considered vaccine hesitant, or if unknown, what percent do not get vaccinated. 

Response: Thank you for this valuable suggestion. We added information about the vaccine hesitancy in the US population in general for the flu vaccine. We avoided the data for the COVID-19 hsitency due to the political controversy.

Is not getting vaccinated the same as vaccine hesitancy?  A better explanation is needed for the reader here with a definition of vaccine hesitancy and a reference.  

Response: Response: Thank you for this valuable recommendation, We added a definition of vaccine hesitancy and information from the World Health Organization.

Line 38-39 has an extra comma

Sentence 39 makes this point better about vaccine hesitancy, and might make a good starting sentence if it included a definition, some statistics, and a reference.

Response: Thank you for this valuable suggestion. We amended the text.

Line 50

When was the HPV vaccine coverage expanded to 45 years compared to its availability in children?  What are the specific ACIP recommendations for this vaccine in adults?

Did coverage for adults happen during the pandemic when people were not getting well visits in person? While it is mentioned in the discussion that intention to vaccinate may have increased, vaccination rates in general could tend to decline during the pandemic, particularly if it is a multi-vaccine series administered in person.

Response: Thank you for this recommendation. We added information regarding the change in guidelines that extended the HPV vaccine to adults. However, there is no data to provide information about the HPV vaccination status during the COVID-19 pandemic.

Line 204

Indentation is off.

Response: Thank you. We amended the text.

Methods

I do not see that the time period of result collection is mentioned, although it looks like it was IRB approved in November 2021.  It is difficult to relate influence of the COVID-19 pandemic to the participant comments when it isn’t clearly stated that it is presumably after COVID-19 vaccines became available.  

Response: Thank you for this suggestion. We added more information in the methods.

It would also be interesting to know why specifically people in Tennessee were chosen.  Are there any known aspects about vaccination rates in Tennessee that would make this population of interest?  For example, average vaccination rates, high vaccination rates, low vaccination rates compared to other parts of the country.

Response: Thank you or this recommendation. This study was sponsored for TN, which has a heterogenous population depending on the geographic location with lower rate of vaccination rates versus other parts of the country.

Results

While the workload of this study seemed labor intensive, participant demographics are very general.  I am left wondering more about who these people are to know who their responses might apply to.  For example, any of the following would be helpful:

  • Have they themselves been vaccinated?  Response:Our inclusion and exclusion criteria for the qualitative study did not ask for vaccination status.
  • Are they sexually active?  Monogamous?  Married? 

Response: We value your suggestion. However, this was a qualitative study where the participants would not be comfortable sharing their private information. Therefore, the questions for this study were reviewed by the IRB.

  • Do they have children who they have agreed to have vaccinated?  
  • What is their level of education?  Some are of college age, are they in college?  
  • Going a step further, have they had HPV?  Has their doctor recommended HPV vaccine for them?

Response: Thank you for this suggestion. Some of the participants stated they were vaccinated.  

  • Are they people who have received other vaccines (e.g., influenza, COVID, etc.?). 

Response: Thank you for this clarification. Most of the participants discussed the COVID-19 vaccination status; however, the scope of this study was to obtain information about HPV and no other vaccines.

Overall, it is very difficult to tell who these general results apply to.  

Also, for men vs. women, women might be more likely to see a doctor (gynecologist) for routine care whereas men ages 18-45 might not be seeing a doctor for routine care.  None of this information is provided.  

Response: Thank you for this comment. It would be inappropriate to generalize our results in the same way that a survey might. Qualitative research is hypothesis generating – and so rather than try to apply our results to “groups” of the population, it’s intent really is to provide depth and breadth of new data on the topic of HPV vaccination in this older population. From there, future research can be conducted over survey, etc. to determine how these themes generalize in a larger population.

As a clinician reading the results, I do not find them helpful.  They are detailed for the sake of completeness, but it is tedious and time consuming to read through the individual’s exact words without a summary table of what the gist of the content is, or what can be done with this information.  

Perhaps the participants’ exact words could be put in a supplemental section or offset from the main text so the reader could skip over if they don’t want to read the comments verbatim.

Response: Thank you for this suggestion. However, we followed the qualitative research guidelines for reporting. We must use verbatim quotes.

The results read as if they would be of interest to industry to sell a product as opposed to being useful to a clinician recommending a vaccine for a patient.

Response: We value your comment. We feel that the identification of the themes from this qualitative study can help clinicians have a better idea of the reason(s) for their patients’ vaccine hesitancy. Provider recommendations are a strong positive factor for vaccination, and this is not done consistently per other studies. These are missed opportunities.

Discussion

The discussion nicely summarizes the results and puts them in context.  A few points that could be considered mentioning to put the results in additional context:

  1. More information on demographics of who the people surveyed are and/or who their opinions might represent (see above).

Response: Thank you for your comment. We did not provide a survey – instead this was a qualitative research study using recorded focus groups. Demographic information that was collected as part of the interviews has been shared in Table 1.

  1. Information on why this broad age range was chosen, aside from the vaccine being approved for this age group.  

Response: This was a series of focus groups and so recruitment strategy was based on creating a stratified sample across regions, ages, and races.

  1. From a public health perspective, it would be interesting to know the opinions of the group at highest risk for new infections that could be prevented by the vaccine.

Response: Thank you for this perspective – we agree that would make for an interesting study. Our aim was specifically in this newly FDA approved age group which is now eligible for the vaccines.

  1. Are there HPV messaging campaigns?  What information already exists that these participants may have access to?  It mentions that HPV vaccines are ‘promoted and viewed’ as female vaccines.  I wonder how the vaccines might have been promoted and how people have access to the information.

Response: Thank you for these questions. The participants did capture what the authors believe is the current US approach to promoting HPV vaccines (although this was not required for them to capture it, nor was it necessary for our research aim). What they did capture was what, of the available promotional material, they recall being exposed to. This was articulated primarily in Theme 3: Vaccine promotion strategies.

The final paragraph is a nice summary of the results and their importance.

Round 2

Reviewer 3 Report

I appreciate the response to comments, but minimal changes we made to the actual manuscript.

Author Response

We have added further details to the limitation section.